**Data Availability Statement:** All relevant data are available from https://neurovault.org/collections/12006/ and the supporting information files.

# A callosal biomarker of behavioral intervention outcomes for autism spectrum disorder? A case-control feasibility study with diffusion tensor imaging

Javier Virues-Ortega[1,2]*, Nicole S. McKay[3], Jessica C. McCormack[4], Nerea Lopez[5], Rosalie Liu[1], Ian Kirk[1]

1 School of Psychology, The University of Auckland, Auckland, New Zealand, 2 Facultad de Psychology, Universidad Autónoma de Madrid, Madrid, Spain, 3 Department of Neurology, Washington University, St Louis, Missouri, United States of America, 4 National Institute for Health Innovation, School of Population Health, University of Auckland, Auckland, New Zealand, 5 Facultad de Psicología, Universidad Nacional de Educación a Distancia, Madrid, Spain

* javier.virues-ortega@uam.es

## Abstract

Tentative results from feasibility analyses are critical for planning future randomized control trials (RCTs) in the emerging field of neural biomarkers of behavioral interventions. The current feasibility study used MRI-derived diffusion imaging data to investigate whether it would be possible to identify neural biomarkers of a behavioral intervention among people diagnosed with autism spectrum disorder (ASD). The corpus callosum has been linked to cognitive processing and callosal abnormalities have been previously found in people diagnosed with ASD. We used a case-control design to evaluate the association between the type of intervention people diagnosed with ASD had previously received and their current white matter integrity in the corpus callosum. Twenty-six children and adolescents with ASD, with and without a history of parent-managed behavioral intervention, underwent an MRI scan with a diffusion data acquisition sequence. We conducted tract-based spatial statistics and a region of interest analysis. The fractional anisotropy values (believed to indicate white matter integrity) in the posterior corpus callosum was significantly different across cases (exposed to parent-managed behavioral intervention) and controls (not exposed to parent-managed behavioral intervention). The effect was modulated by the intensity of the behavioral intervention according to a dose-response relationship. The current feasibility case-control study provides the basis for estimating the statistical power required for future RCTs in this field. In addition, the study demonstrated the effectiveness of purposely-developed motion control protocols and helped to identify regions of interest candidates. Potential clinical applications of diffusion tensor imaging in the evaluation of treatment outcomes in ASD are discussed.

**Funding:** This study was supported by grants from Oakley Mental Health Foundation (project no. 3705925), The University of Auckland (Marsden Fund Near Miss Support Program), and through a research contract between ABA España and The University of Auckland (project no. CON02739); all awarded to JVO.

**Competing interests:** The authors have declared that no competing interests exist.

## Introduction

Children with autism spectrum disorder (ASD) present with learning delays in social behavior and language, and often engage in stereotypic behavior. In the absence of intensive educational and psychosocial interventions, children with ASD follow a non-remission course often leading to disability and dependence during adulthood. The lifetime cost of an individual with ASD has been estimated at $1.4M in the US and £1.2M in the UK, with these figures almost doubling when there is comorbid intellectual disability [1]. Special education services and parental productivity loss make up most of this cost during childhood, while residential services and individual productivity loss are the main contributors during adulthood.

Unlike other neurodevelopmental disorders, the causal mechanisms of autism are not well understood at the molecular, cellular or system level. In addition, there is some debate regarding the extent to which some on the autism spectrum have a "deficit" per se [2]. Nevertheless, children on the autism spectrum tend to undergo an atypical brain maturation resulting in neuroanatomical and functional differences relative to neurotypical children. Neuroimaging studies have repeatedly reported an atypical connectivity among children with autism consistent with reduced communication between distant brain regions [3, 4]. Studies have also shown an altered functional connectivity when participants engage in a variety of cognitive tasks and also during resting-state analyses (see a review in [5]).

In recent years, the use of diffusion tension imaging (DTI) has allowed novel insights on the macrostructure and microstructure of white matter (WM) in people with autism. This technique examines the WM anisotropic water diffusion as informed by fractional anisotropy (FA) and other diffusivity metrics [6]. Fractional anisotropy, in particular, is a composite value related to axonal density, size, myelination, and fiber organization [7], and provides an indication of structural configuration and brain connectivity.

Studies exploring DTI among people with ASD have shown an altered FA in several WM tracts spanning across different areas of the brain. The most consistent findings have been reported for the corpus callosum, cingulum, uncinate fasciculus, arcuate fasciculus, and the superior and inferior longitudinal fasciculi [3, 4]. Atypical WM architecture of the corpus callosum is among the most consistently reported DTI findings in people with autism [8].

The callosal commissure is the largest interhemispheric WM bundle and it is thought to be involved in social functioning [9], motor skills [9–12], and complex cognitive repertoires [12–14]. The corpus callosum has become a focus of interest for neuroscientific research in autism. Decreases in the volume of the corpus callosum have been reported in several areas including the forceps major (splenium), forceps minor (genu), and corpus callosum body [15]. Interestingly, behavioral similarities have been reported among people with autism and those with callosal agenesis [4, 16]. These include deficits in social skills, problem solving, and abstract reasoning [9, 17].

Neural changes have been documented in a variety of populations due to motor learning and practice [18], physical exercise [19], memory training [20], reading intervention [21], and cognitive therapy [22], to mention a few. Within the ASD population, a seminal study by Pardini et al. [23] reported a relation between WM integrity in the uncinate fasciculus and the duration of cognitive and behavioral treatments.

Comprehensive meta-analyses have shown that both clinic-based and parent-managed intensive behavioral intervention can produce long-term gains in IQ, receptive and productive language, and psychosocial functioning in children with autism [24, 25]. While the outcomes of both parent-managed and clinic-based intensive behavioral intervention for autism are well established [24, 26, 27], there is a dearth of studies evaluating their potential effects on neural plasticity. In particular, few neuroimaging studies have evaluated brain connectivity in relation to comprehensive evidence-based treatments for ASD.

The current feasibility analysis presents a case-control study of parent-managed behavioral intervention (PMBI) for autism. A case-control feasibility study present distinct advantages in this research context. First, it can help to identify potential regions of interest and estimate likely effect size ranges, which are critical for planning future RCTs. Second, feasibility studies can help to determine whether more resource-intensive treatment evaluation designs are warranted [28]. The cases included in the study were comprised of children with autism that had been exposed to PMBI, while controls had received other services. We hypothesized that PMBI can modulate changes in WM integrity informed by FA in individuals with autism. Specific hypothesis regarding the affected WM tracts were not held due to the inconclusive evidence available in the literature, except for the corpus callosum, where we hoped to find differences in the WM microstructure integrity. We first conducted a whole-brain exploratory analysis using track-based spatial statistics (TBSS). This initial approach allowed us to examine whether the brain was globally impacted by ASD treatment exposure status and helped to inform a subsequent region of interest (ROI) analysis. We hypothesized that a history of PMBI may modulate FA in the corpus callosum in children and adolescents with ASD.

## Methods

### Participants

Participants were recruited thorough a news release on a national newspaper, and a mailing campaign through an autism support network. In order to be admitted into the study, participants required (a) a clinical diagnosis conducted by a multidisciplinary team often lead by a pediatrician, child psychiatrist or clinical psychologist using the diagnostic criteria of the 4th or 5th edition of the Diagnostic and Statistical Manual of Mental Disorders (DSM) [29, 30], and (b) no presence of any metallic objects or fragments that would exclude them from participation in an MRI machine. The diagnostic process was often supported by the administration of standardized assessments including the Autism Diagnostic Observation Schedule (ADOS) and the Autism Diagnostic Interview–Revised (ADI-R).

A total of 42 participants from the Auckland region and neighboring rural areas in New Zealand expressed interest in the study and met the inclusion criteria. From this pool of participants, 30 individuals were sequentially invited to undergo an MRI scan. One of the subjects could not attend the appointment, another three were removed from the analysis due to excessive head motion. The final sample of 26 individuals (23 males and 3 females, mean age: 13.81 ± 5.04) included 19 subjects diagnosed with ASD, four with Asperger syndrome, and three with pervasive developmental disorder not otherwise specified (PDD-NOS). Six of these participants had comorbid diagnoses of ASD and ADHD, one had comorbid diagnoses of Asperger syndrome and ADHD, and one had comorbid diagnoses of PDD-NOS and ADHD. In addition, 11 participants had a comorbid diagnosis of intellectual disability (Table 1).

Cases were defined as individuals whose direct caregiver had received parent training for the purposes of conducting PMBI. The caregivers of participants exposed to PMBI had received instruction and supervision by a behavioral consultant on educational strategies, whether as a stand-alone intervention or in the context of early-intensive behavioral intervention (EIBI) based on applied behavior analysis. Caregivers had received training in areas including daily living skills, language and communication, leisure and social behavior, and academic abilities. Controls were individuals whose direct caregivers had not received behavioral parent training as defined above. Overall, 13 participants received PMBI, while 13 had received other services. Case and controls were compared in a range of personal and clinical characteristics. These included sex, age, ethnicity, primary diagnosis, ADHD and intellectual disability comorbidities, DSM severity (i.e., level of support required), changes in severity

**Table 1. Participant characteristics.**

| | PMBI (*n* = 13) | Other (*n* = 13) | *p* value |
|---|---|---|---|
| Sex (males)[1] | 100.0 (13) | 69.2 (9) | .033 |
| Age (years)[2] | 12.51±4.65, 6.50–23.08 | 15.11±5.24, 7.68–23.98 | .234 |
| Ethnicity | | | .793 |
| Caucasian | 69.2 (9) | 76.9 (10) | |
| Asian | 7.7 (1) | 15.4 (2) | |
| Maori | 15.4 (2) | 0.0 (0) | |
| Other | 7.7 (1) | 7.7 (1) | |
| ASD diagnosis | | | .344 |
| Autism | 76.9 (10) | 69.2 (9) | |
| Asperger syndrome | 15.4 (2) | 15.4 (2) | |
| PDD-NOS | 7.7 (1) | 15.4 (2) | |
| Selected comorbidity | | | |
| Intellectual disability | 23.1 (3) | 61.5 (8) | .052 |
| ADHD | 38.5 (5) | 23.1 (3) | .193 |
| Autism symptoms | | | |
| When first diagnosed | 13.18 ± 3.52, 6–18 | 15.09 ± 2.74, 10–18 | .171 |
| Currently | 6.33 ± 2.77, 1–11 | 9.50 ± 2.58, 6–14 | .008 |
| DSM severity | 1.23 ± 0.44, 1–2 | 1.54 ± 0.52, 1–2 | .225 |
| Requires support | 76.9 (10) | 46.2 (6) | |
| Substantial support | 23.1 (3) | 53.8 (7) | |
| Severity differential | -0.92 ± 0.86, -2–0 | -0.92 ± 0.86, -2–0 | .840 |
| Mainstreamness | 2.62 ± 0.87, 0–3 | 2.38 ± 0.96, 0–3 | .527 |
| Home-schooled | 7.7 (1) | 7.7 (1) | |
| Special education school | 0.0 (0) | 7.7 (1) | |
| Special education classroom | 15.4 (2) | 23.1 (3) | |
| Mainstream | 81.8 (10) | 61.5 (8) | |
| Current level of support | | | |
| Daily special education hours | 2.23 ± 1.42, 0–5 | 2.58 ± 1.17, 1–5 | .969 |
| Weekly teacher aid hours | 10.00 ± 10.48, 0–30 | 12.15 ± 13.26, 0–30 | .840 |
| Interventions (total) | 5.54 ± 1.66, 3–8 | 3.69 ± 1.48, 2–7 | .006 |
| Sensory integration | 15.4 (2) | 15.4 (2) | |
| Dietary interventions | 76.9 (10) | 23.1 (3) | |
| Occupational therapy | 30.8 (4) | 30.8 (4) | |
| CBT | 23.1 (3) | 15.4 (2) | |
| SLT | 46.2 (6) | 38.5 (5) | |
| Social worker | 15.4 (2) | 15.4 (2) | |
| Social support group | 23.1 (3) | 30.8 (4) | |
| Equine-assisted therapy | 15.4 (2) | 15.4 (2) | |
| Early intervention (non EIBI) | 15.4 (2) | 15.4 (2) | |
| Medical | 38.5 (5) | 15.4 (2) | |
| Other therapies or services | 23.1 (3) | 53.8 (7) | |

*Notes*. 1. % (*n*); 2. *Mean ± SD*, range. ANOVAs or non-parametric tests, as appropriate. Critical *p* value according to Benjamini and Hochberg [31] multiple-comparison correction is .005. Ad hoc autism severity questionnaire included in S1 Appendix. Severity differential was calculated as the difference in DSM-defined severity when first diagnosed and at the time of the study. Mainstreamness defined as the average ordinal level of the New Zealand Ministry of Education Classification (0 = Homeschool/correspondence, 1 = Special education school, 2 = Special education classroom, 3 = Mainstream). ADHD = Attention deficit and hyperactivity disorder; ASD = Autism spectrum disorder; CBT = Cognitive behavioral therapy; DSM = Diagnostic and Statistical Manual of Mental Disorders; EIBI = Early intensive behavioral intervention; PDD-NOS = Pervasive developmental disorder not otherwise specified; PMBI = Parent-managed behavioral intervention; SLT = Speech language therapy.

defined as the difference in DSM severity when first diagnosed and at the time of the study, level of mainstream school integration (i.e., home-schooled, special education school, special education classroom, mainstream school), number of daily special education hours received at school, and number of teacher aid hours per week. In addition to PMBI, we documented all current and historical interventions participants had received according to the following categories: sensory integration therapy, dietary interventions (including gluten-free diets), occupational therapy, cognitive-behavioral therapy, speech-language therapy, social worker support, autism social support group (regular attendance to parent groups or other autism-related activities), equine-assisted therapy, early intervention programs (different from EIBI), medical and pharmacological interventions, and other therapies or services. All participant characteristics including treatment history were informed by the primary caregiver by way of a structured interview. Participants did not differ significantly in sociodemographic or treatment history characteristics (there were statistical trends, after multiple-comparison correction, for sex, intellectual disability comorbidity, and total number of interventions received, see Table 1). Interestingly, those in the PMBI group showed a trend toward lower autism symptoms at the time of the study, but not when first diagnosed.

The current study was approved by the University of Auckland's Human Participants Ethics Committee (Ref: 014993). Parents provided informed consent in writing. All participants also provided assent to the study procedures.

## Mock scanner training

All participants underwent a minimum of one and a maximum of two mock scanner sessions to become accustomed to the neuroimaging procedure. During these sessions, participants underwent an abbreviated stillness training procedure developed by Cox et al. [32]. This procedure involved evidence-based behavior modification procedures including prompting, stimulus fading, and contingent social reinforcement as a means to minimize participant's head and body movement during the mock scanner sessions.

## Image acquisition

Imaging data was acquired with a 1.5 Tesla Siemens Avanto scanner (Erlangen, Germany) both for MRI and DTI. T1 MRI images were acquired using a MPRAGE sequence with a voxel resolution of 1x1x1 mm. The voxel size for DTI was 2 x 2 x 2 mm. We used a single shot spin-echo echo planar imaging (EPI) sequence (repetition time = 9746; echo time = 101; field of view = 256; matrix size = $256 \times 256$) applied in 12 non-collinear directions. Three runs were collected to allow for averaging to occur along each diffusion direction and improve estimations of diffusion indices. Images were acquired with a diffusion weighting of b = 1000 s/mm$^2$, and a reference image with a diffusion weighting of b = 0 s/mm$^2$ was also collected. The total acquisition time was approximately 10 minutes.

## DTI processing

White matter microstructure integrity was compared between participants with ASD that had or had not received PMBI. We conducted a whole-brain voxel-based comparison with TBSS. The TBSS allowed us to focus on the major tracts that are broadly believed to be affected in people with ASD (e.g., corpus callosum). Image files were transferred into a Linux work station for processing. DICOM files were converted to NIFTI using MRICRO. Preprocessing was conducted using the FSL FMRIB's Diffusion Toolbox (FDT) (FSL 5.0; www.fmrib.ox.ac.uk/fsl). Quality assurance involved eddy current induced distortion and head motion correction, as well as removal of non-brain tissue. Given that motion introduces artifacts in DTI metrics,

only data with a mean absolute RMS less than 5 mm was included and image quality passing visual inspection (refer to participant attrition above). A binary mask of the brain was generated for each subject from their no-diffusion image. We then used FDTIFIT to fit the diffusion tensor model. We generated FA maps for each participant.

## TBSS analysis

As an exploratory analysis, we first began with a whole brain approach using TBSS. To remove likely outliers, TBSS first erodes the FA images of each subject slightly, and zeros end slices. Nonlinear registration to a common space was conducted using the FNIRT tool, which aligns all subjects' FA images to the 1 x 1 x 1 mm FMRIB58_FA standard-space target, namely, the most representative FA image identified. Each subject's image is also affine transformed to MNI152 space resulting in a standard-space version of each subject's FA image. Next, an image detailing the mean FA for all subjects was created and thinned to create a mean FA skeleton, which represents the centers of all tracts common to the group. The FA data for each subject were then projected onto the skeleton. Variance from the FA maps related to the age and sex factors was removed (intracranial volume was not significantly different across groups, $t = -0.78$, $p = 0.44$). The resulting images were then subjected to TBSS within FSL. FSL's *randomise* function with a threshold-free-cluster-enhancement method was applied for the group-level variable. Final *t*-stat images are displayed in Fig 1 (a full image collection is available in https://neurovault.org/collections/12006/). While our main interest for the current exploratory analysis was FA, which is a general indicator of WM integrity, we replicated the procedure for mean diffusivity (MD), axial diffusivity (AD), and radial diffusivity (RD).

## Tractography analysis

We conducted a targeted tractography analysis focusing on FA in the corpus callosum. Other tracts that the TBSS analysis showed to be significantly different, for FA and other diffusivity metrics are reported in the S1 Dataset for context. Thereby, the corpus callosum, superior longitudinal fasciculi, cingulum and the uncinated fasciculi were reconstructed. We used the preset parameters of the FreeSurfer Software Suite (v. 6.0.0) for the selected tracts mentioned above. In the generated connectivity map each voxel represents a connectivity value where the higher the number, the greater the probability of the pathway passing through that voxel. Due to the non-specific and widespread distribution of connections across the brain, the reconstruction of the tracts was performed with an FA 0.55 to ensure only the inclusion of major tracts with high probability and to avoid false positives and pathways that may be the result of image noise. The dissection of these tracts was performed according to the procedure described by Catani and Thiebaut de Schotten [33]. The thresholded connectivity maps were binarized and masked into the FA map to derive mean values for each of the focused regions. The same procedure was repeated for MD, AD, and RD metrics.

Statistical comparisons of the tractography outcome measures were performed using the statistical package SPSS Statistics v. 27 (IBM, Armonk, New York, United States). We computed quantile-quantile plots and skewness analyses to verify that the distribution of the dependent variables did not present notable departures from distribution normality and symmetry, respectively. We used the FA of two tracts of interest (forceps major and forceps minor) as dependent variables and exposure status as grouping variable (i.e., exposed to PMBI, not exposed to PMBI). We expected that a history of PMBI would modulate FA in the corpus callosum. We conducted univariate analyses of variance (ANOVA) to compare FA of selected WM tracts across cases and controls (Model 1). Due to sample size restrictions, covariate-specific analyses were not conducted. However, descriptive variables that had shown a statistical

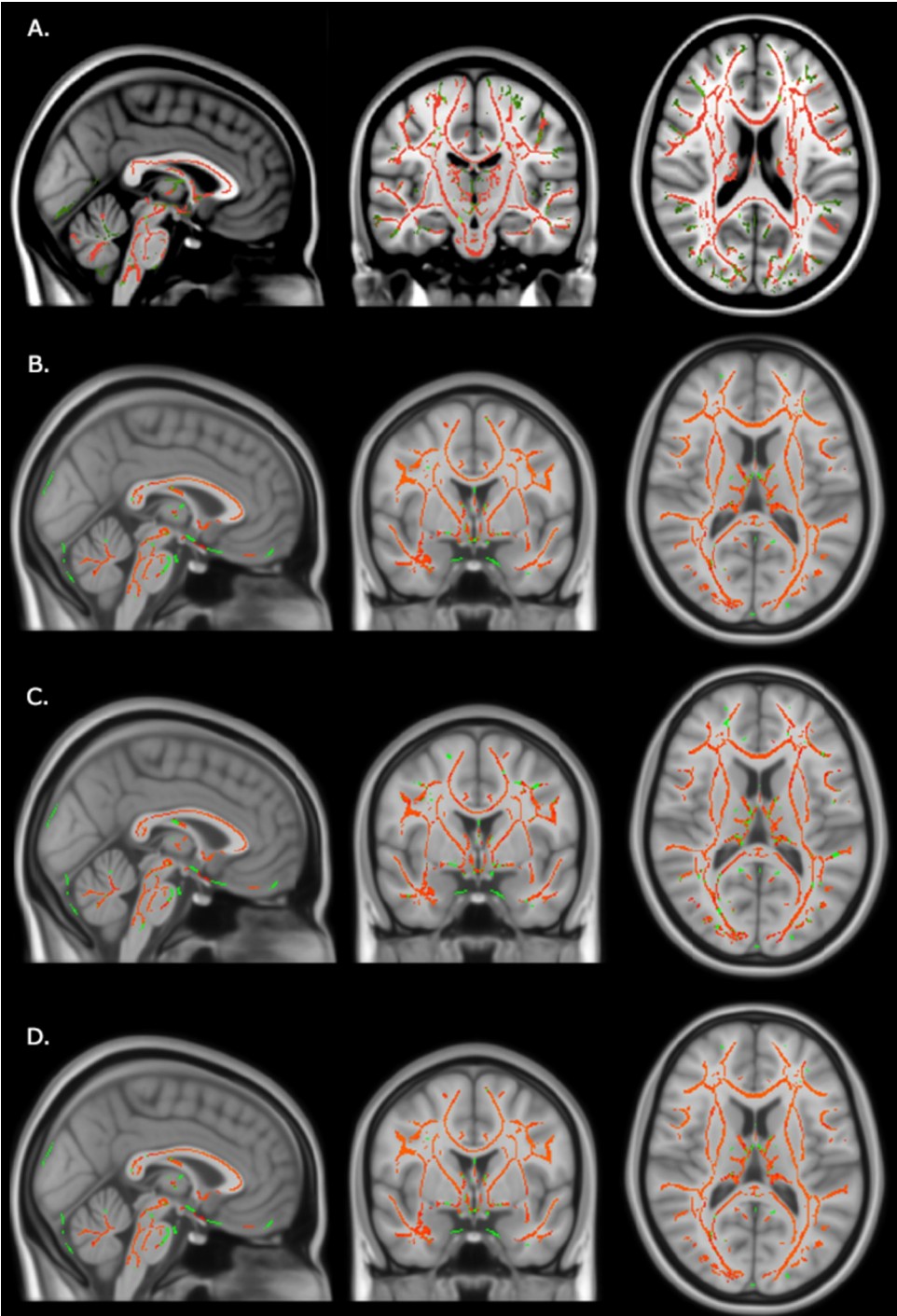

**Fig 1.** Red voxels denote greater fractional anisotropy (A), mean diffusivity (B), axial diffusivity (C), and radial diffusivity (D) among individuals exposed to parent-managed behavioral intervention ($n = 13$) relative to individuals not exposed to parent-managed behavioral intervention ($n = 13$). Green voxels indicate the mean WM skeleton of all subjects. Red and green voxels are plotted onto longitudinal, sagittal and horizontal standardized anatomical images.

trend were selected to be included in a corrected ANOVA as covariates (Model 2). We computed partial eta squared ($\eta_p^2$) effect sizes for all ANOVAs [34]. In order to provide an indication of causality, we conducted dose-response analysis by exposure intensity. Specifically,

**Table 2. Fractional anisotropy in the corpus callosum among cases (n = 13) and controls (n = 13).**

| | M (SD) | | F | p | Critical p[a] | $\eta_p^2$ |
|---|---|---|---|---|---|---|
| | **Cases** | **Controls** | | | | |
| Model 1 | | | | | | |
| Forceps major | 0.58 (0.06) | 0.62 (0.04) | 3.63 | .069 | .025 | .13 |
| Forceps minor | 0.46 (0.05) | 0.49 (0.07) | 1.80 | .193 | .025 | .07 |
| Model 2 | | | | | | |
| Forceps major | 0.58 (0.06) | 0.62 (0.04) | 7.83 | .011 | .025 | .28 |
| Forceps minor | 0.46 (0.05) | 0.49 (0.07) | 0.83 | .374 | .025 | .04 |

*Notes.* All univariate ANOVAs. a. Critical *p* according to Benjamini and Hochberg [31] multiple-comparison correction. Model 2 includes age, sex, intracranial volume, intellectual disability comorbidity, and total number of interventions as covariates.

individuals with low intensity behavioral intervention were those that had not received PMBI (controls). Individuals with medium intensity PMBI where those exposed to parent training *without* EIBI. Finally, individuals with high intensity behavioral intervention were those exposed to parent training in addition to (or in the context of) EIBI. The dose-response analysis was conducted by way of a univariate ANOVA with treatment intensity status as predictor variable (low, medium, high) and FA in the corpus callosum as outcome.

## Results

The results of the voxelwise TBSS analysis showed significant differences between the WM microstructure of children with autism that had received PMBI when compared to those who did not. Individuals exposed to PMBI revealed multiple regions with increased FA, MD, AD, and RD within WM pathways, which included the corpus callosum, superior longitudinal fasciculus, uncinate fasciculus, and the cingulum. This finding was replicated for other (Fig 1).

The subsequent ROI analysis focused on the corpus callosum (Table 2). We found a significantly different WM diffusion in the posterior region of the corpus callosum among participants exposed to PMBI, $F(1,25) = 7.83$, $p = .011$, $\eta_p^2 = 0.28$. This result was specific to the posterior portion of the corpus callosum or forceps major. Specifically, the corpus callosum of children exposed to PMBI did not display differences in FA in the body and most anterior sections of the corpus callosum (forceps minor) relative to the controls, $F(1,25) = 0.83$, $p > .1$, $\eta_p^2 = 0.04$. The effect was evident in the corrected model (Model 2), which included age, sex, intracranial volume, intellectual disability comorbidity, and total number of interventions as covariates (Table 2). The dose-response analysis showed a gradual effect of intervention intensity on FA in the forceps major, $F(2,25) = 4.47$, $p = .026$, $\eta_p^2 = 0.32$ (see Table 3). The statistical

**Table 3. Parent training intensity and functional anisotropy in the forceps major.**

| Intensity | M (SD), n | F | df | p | $\eta_p^2$ |
|---|---|---|---|---|---|
| Low | 0.62 (0.04), 13 | 4.47 | 2 | .026 | .32 |
| Medium | 0.60 (0.05), 5 | | | | |
| High | 0.57 (0.07), 8 | | | | |

*Notes.* Univariate ANOVA with total number of treatments, age, sex, intracranial volume, intellectual disability comorbidity, and total number of interventions as covariates. Low intensity = no parent training reported; Medium intensity = parent training without early intensive behavioral intervention; High intensity = parent training in addition to (or in the context of) early intensive behavioral intervention.

analyses did not establish any WM tracts of interest as significantly different across cases and controls for any of the other DTI metrics (MD, AD, and RD).

## Discussion

The present feasibility study expands on previous DTI analyses conducted with children with ASD. The existing literature has largely focused on abnormal tract development that may be specific to those receiving the diagnosis of ASD. However, there is a dearth of studies evaluating the potential impact of varying psychosocial treatment histories on WM microstructure and functioning. Several independent randomized and non-randomized trials support EIBI as an evidence-based remedial approach for the cognitive, verbal and social deficits of children with autism [35] and the intervention is now considered standard practice by a number of authoritative sources (see for example [36]). A meta-analysis by Virues-Ortega [25] showed that PMBI programs produce essentially identical effect sizes relative to clinic-based programs in all outcomes evaluated including IQ, non-verbal IQ, receptive and expressive language, and adaptive behavior.

We have used TBSS analysis and tractographic methods to assess the WM integrity in children with ASD. As an exploratory analysis, we first began with a whole-brain TBSS analysis, which was run for each grouping to ascertain the extent to which the brain of children with autism was influenced by their history of PMBI. The results showed that exposed individuals had higher FA in regions such as the corpus callosum, superior longitudinal fasciculi, left and right cingulum, and uncinated fasciculi. We then conducted a targeted ROI analysis focusing on the corpus callosum identified in the TBSS analysis. The posterior region of the corpus callosum, the forceps major, was found to have significantly lower FA among those exposed to PMBI. The effect was found to follow a dose-response relation when the intensity of PMBI was used as a predictive variable.

Our tentative results are consistent with the view that it may be possible for ontogenic exposures such as PMBI to exert a long-term influence on the neurophysiology of children and adolescents with ASD. Analyses such as the one presented here may help to identify neurophysiological biomarkers of treatment outcomes in future RCTs with larger sample sizes. While the corpus callosum volume has been suggested as a biomarker of autism [37], it has not yet been proposed that a specific region of the corpus callosum may be a candidate marker of exposure to various treatment histories. The search for biomarkers of treatment outcomes in autism remains a largely understudied area. In a notable exception, Bradshaw et al. [38] showed that six months of behavioral intervention (i.e., pivotal response treatment [39]) induced verifiable changes in eye motion toward social stimuli. However, neurophysiological biomarkers are yet to be established.

Interestingly, our results provide additional context to the literature that has identified the corpus callosum as an important brain structure for those with autism. For example, Haar et al. [40] assessed anatomical MRIs of over 1,000 individuals from the Autism Brain Image Data Exchange project. The authors reported that individuals with autism had lower corpus callosum volumes relative to age-matched typically developing peers. The authors had divided the corpus callosum in five segments along the anterior-posterior axis for their analysis. Only the central segment produced a mild albeit statistically significant effect size ($d = 0.2$). Moreover, when analyzed individually, only two of the 18 participating sites showed the effect. A meta-analysis of DTI studies in autism has also identified the corpus callosum (and the splenium in particular) as the location of significant FA alterations [8].

The corpus callosum has remained a region of interest in autism for some time in the context of brain connectivity and synchronization theories of autism [41]. Interestingly, an earlier

meta-analysis by Frazier and Hardam [37] summarizing 10 studies with a pooled sample of 253 individuals with autism had reported relatively large differences in the corpus callosum although the effect disappeared caudally. In this connection, the involvement of the posterior segment of the corpus callosum reported in the current analysis, if substantiated in subsequent studies, may be a unique phenomenon that may not be assimilated simply to a wider involvement of the corpus callosum in people with autism.

It is necessary to clarify the difference in the results of our analytical methods, since TBSS has found that most of the WM of those exposed to PMBI has higher FA values, whereas ROI analyses have only established a difference in FA in the posterior region of the corpus callosum. This can be explained by the different approach of these two methods. Specifically, TBSS and ROI analyses test different aspects of the WM pathways. By focusing only on the voxels of each path that are present in every subject, and not the entire tract of each individual, TBSS can control for the anatomical differences between subjects. On the other hand, ROI analyses test FA across all voxels within the tract for each individual, thereby being more vulnerable to WM covariates as age or comorbid disorders. Thus, it is possible that the age range and existing comorbidities may have had an impact on our ROI analysis (see for example [42]). We have mitigated this concern to the extent possible by conducting a thorough comparison of numerous descriptive variables and adding critical covariates to our analytical models.

Since the present study was not an RCT but a feasibility case-control study with a relatively small sample, our results should be considered in light of some limitations. First, the lack of a control group should lead to caution in the interpretation of these results. Participants that were exposed or not exposed to PMBI were comparable in a range of critical characteristics. Future longitudinal RCTs should prospectively compare a non-PMBI control group with a PMBI intervention group before the intervention and at various time points over the course, and, potentially, after the intervention.

The age of participants may be an important cofound. While total brain volume remains relatively constant after age five, internal remodeling occurs within the brain. For example, Mills et al. [43] examined a large longitudinal sample (ages 8–30 years) finding that cerebral white mater increases gradually from childhood until mid-to-late adolescence. While cases and controls did not differ significantly in age or intracranial volume, it would be beneficial in future studies to shorten the age range of participants to minimize any age-mediated volume variability.

The current sample of participants reflects the sex distribution of the autistic population. Therefore, cases and controls were not sex-matched. Subgroup analysis by sex and other critical characteristics including pre-intervention functioning, treatment duration, and treatment success will require larger samples in order to highlight WM microstructural differences in subgroup analyses.

From a methodological standpoint, TBSS attempts to overcome the shortcomings of voxel-based morphometry and ROI analyses. However, it remains a concern that TBSS does not account for head motion within the scan. While the mock scanner procedure and data pre-processing minimize the effects of head movement, these could cause false FA values to be reported. It is important to indicate that the current data were collected with only 12 motion probing gradients. While six directions have been theorized to be sufficient for diffusion-weighted analyses focusing on FA differences, current diffusion study protocols usually employ 30 directions or more. Therefore, increasing the number of motion probing gradients would improve the resolution of the scans, but it is unlikely that our results would have been significantly skewed because of a lack of scanning directions. Additionally, our protocol consisted of three runs, allowing us to average across each gradient direction and improve our ability to estimate diffusion indices.

The current study aimed to explore relationships between common interventions of ASD and WM integrity. In spite of our tentative findings, it is important to highlight that FA is known to reflect a variety of WM changes. For example, despite being commonly associated with decreased tract integrity, increased FA could also reflect an increase in neurons, an increase in myelin, or increased inflammation. In order to better characterize what biological mechanism is underpinning the changes we are observing in these brains, future extensions of this work would need to consider mean diffusivity, axial diffusivity, and radial diffusivity in greater detail that it had been possible with the current dataset.

Finally, the results of our study could be strengthened by the application of DTI techniques at the beginning of the therapeutic process to better characterize the causality link between DTI metrics and therapy. Likewise, the case-control design did not allow for greater homogeneity among those exposed and not exposed to PMBI in terms of their treatment histories. The presence of baseline DTI data and pre-specified treatment integrity criteria could help to verify if there is a link between pre-treatment mean FA values and treatment effectiveness.

A potential extension of the current feasibility study would involve to replicate the proposed design, including the dose-response analysis, within a large neuroimaging repository. Unfortunately, existing databases, including the Autism Brain Imaging Data Exchange (ABIDE I and ABIDE II) [44], do not include treatment data. The inclusion of treatment outcome data would be a positive addition to these collections, maybe following international guidelines for ASD treatment outcomes (see, for example, ICHOM Connect [45]). Larger samples sizes with narrower age ranges and longitudinal analyses with matched controls or randomized group assignment are desirable methodological standards for future research in this area. To our knowledge, relations between treatment efficacy and DTI measures have not yet been reported in the ASD population, making this an extremely important avenue for future research. In addition, subgroup analyses by age can help to determine whether early intensive interventions could have a long-lasting impact on brain development later in childhood and into the adolescence and adult age.

## Conclusions

The current feasibility study used MRI-derived diffusion imaging data (TBSS and seed-based tractography) to investigate whether there was a relationship between the intervention received by individuals diagnosed with ASD and their current brain connectivity. In particular, we report differences in the WM integrity of the posterior corpus callosum in those exposed to PMBI. This preliminary finding was substantiated by a PMBI intensity dose-response analysis. The corpus callosum is the largest interhemispheric WM bundle and has been previously associated with functional and structural abnormalities in people diagnosed with autism, being an important target area for future analyses. The preliminary results are consistent with disorder-specific alterations of the WM microstructure in people with ASD and is the first to apply neuroimaging techniques to determine whether there is a relationship between intervention history and current brain connectivity. The study also demonstrated that a purposely-developed behavioral protocol for motion control can be used effectively to obtain usable neuroimaging with minimal experimental mortality. Therefore, the present case-control feasibility study provides the basis for more resource-intensive treatment evaluations including RCTs, and longitudinal RCTs in particular, to be conducted in the future. The current line of work will help to explore clinical applications of DTI to measure treatment efficacy in ASD and other neurobehavioral disorders. Progress in the emerging field of neural biomarkers of behavioral interventions may be critical to enhance our understanding of the neural processes mobilized by intensive interventions and to identify early biomarkers of treatment outcomes.

## Supporting information

**S1 Appendix. Ad hoc autism severity questionnaire.**
(PDF)

**S1 Dataset. Full dataset and variable dictionary.**
(XLSX)

## Acknowledgments

The New Zealand Herald (Jamie Morton) and Autism New Zealand (Dane Dougan) assisted with the recruitment process. Maram Abomaray led the implementation of the MRI tolerance protocol with research assistants Monica Widjaja, Margaret Gertzog, Paul Naveen, Zoe Butcher-McGunnigle, and Vinayak Dev. Research assistants also interviewed participating families. Agustin Perez-Bustamante Pereira tabulated a subset of the interview data. We thank Dr. Cemal Koba (University of Trento) and Dr. Andrew Jahn (University of Michigan) for their assistance in the data analysis process, and Dr. Torsten Baldeweg (University College London) for his critical comments and suggestions.

## Author Contributions

**Conceptualization:** Javier Virues-Ortega, Ian Kirk.

**Data curation:** Javier Virues-Ortega, Jessica C. McCormack, Nerea Lopez, Rosalie Liu.

**Formal analysis:** Javier Virues-Ortega, Nicole S. McKay, Nerea Lopez, Rosalie Liu, Ian Kirk.

**Funding acquisition:** Javier Virues-Ortega.

**Investigation:** Javier Virues-Ortega, Jessica C. McCormack.

**Methodology:** Javier Virues-Ortega, Nicole S. McKay.

**Project administration:** Javier Virues-Ortega, Jessica C. McCormack.

**Resources:** Javier Virues-Ortega.

**Software:** Ian Kirk.

**Supervision:** Javier Virues-Ortega, Jessica C. McCormack.

**Writing – original draft:** Javier Virues-Ortega, Nicole S. McKay.

**Writing – review & editing:** Javier Virues-Ortega, Jessica C. McCormack, Ian Kirk.

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
