## [Decision Letter · Decision Letter 0]

20 Oct 2021

PONE-D-21-30729A Callosal Biomarker of Behavioral Intervention Outcomes for Autism? A Case-Control Feasibility Study with Diffusion Tensor ImagingPLOS ONE

Dear Dr. Virues-Ortega,

Thank you for submitting your manuscript to PLOS ONE. After careful consideration, we feel that it has merit but does not fully meet PLOS ONE’s publication criteria as it currently stands. Therefore, we invite you to submit a revised version of the manuscript that addresses the points raised during the review process. I agree with the concern shared by both Reviewers regarding the sample size. The manuscript would be greatly improved by Reviewer 1.Regardless, please also pay carefully attention to the inconsistencies in the sample size/excluded subjects that has been pointed out by the same Reviewer. I also encourage you to explore other DTI-derived metrics, given the highly non-specific nature of FA. Please also be careful to define terms upon first use (e.g., ADOS, ADIS, TBSS [which is also missing an appropriate reference to the original Smith et al. paper). Please also use the term "sex" rather than "gender", the former which is the biological meaning whereas the latter is the sociological construct, which is not relevant in the context of your manuscript. I also note that you used FreeSurfer, but no T1 image is described in the imaging protocol. Finally, please carefully the manuscript for typos (e.g., "Read and green" rather than "Red and green", "mod-to-late" rather than "mid-to-late")

We look forward to receiving your revised manuscript.

Kind regards,

Niels Bergsland

Academic Editor

PLOS ONE

Journal Requirements:

2. Please note that according to our submission guidelines (http://journals.plos.org/plosone/s/submission-guidelines), outmoded terms and potentially stigmatizing labels should be changed to more current, acceptable terminology. To this effect,  “Caucasian” should be changed to “white” or “of [Western] European descent” (as appropriate)

Reviewers' comments:

Reviewer's Responses to Questions

**Comments to the Author**

1. Is the manuscript technically sound, and do the data support the conclusions?

Reviewer #1: No

Reviewer #2: Partly

2. Has the statistical analysis been performed appropriately and rigorously? 

Reviewer #1: No

Reviewer #2: No

3. Have the authors made all data underlying the findings in their manuscript fully available?

Reviewer #1: No

Reviewer #2: Yes

4. Is the manuscript presented in an intelligible fashion and written in standard English?

Reviewer #1: Yes

Reviewer #2: Yes

5. Review Comments to the Author

Reviewer #1: - The provided data repo link (neurovault.org/collections/ZEPLXWYX , dated +2years ago) does NOT contain the input data for the 30 subjects (nor the 26 used)..only final images.

- It was said that 30 subject were recruited, one left study, another 4 removed, and 26 remaining! (in this way you started with 31 subjects, or ended with 25 as the case)...in another place, "26 of the 29 subjects passed quality assurance, discarding 3 individuals" implying only 3 had artifacts, and original were 29. Both sentences needs to be altered to be more clear. following, only the age of males were stated ..why?

- Again, the 26 subjects contained 17 asd, 4Asperger, 4 otherwise (totaling 25?!). It's also useful to identify which seven has ADHD (all ASD? asperger? which mix?) Especially as they are identified as a separate raw on table1 (raising total count to 32?)

- It would be beneficial applying the same procedure on a larger publicly available datasets if possible, maybe a subset of Autism Brain Image Data Exchange or others, to confirm generalizability of findings and mitigate the effects of using a small sample size (n<30) .

- In defining PMBI cases: how long does parent actively conducted those learnt training on their kids? Since this matter directly affect the hypothesis of brain - changes, as some may only conducted those strategies for negligible amount of times. Moreover, what are other services include? and why it's hypothesized it has no effect on the ROI of study, in a way similar to PMBI? Finding an actual distinction can be a result of many other obvious cofounding variables.

- "PMBI can lead to measurable volume changes in known regions of interest for autism": DTI shouldn't reveal volume changes, but micro white matter architecture and integrity.

- Why only FA was investigated? Not also other common measures as mean diffusivity/ axial diffusivity/ ... ?

- The use of references in the discussion doesn't help a lot. For example, Results were said to be consistent with literature, and an example [48] was given although it was anatomical, not WM connectivity (as well as 50).

- Line 143: end of sentence missing (are what?)

Reviewer #2: Virues-Otega et al. evaluated white matter integrity between subjects with autism spectrum disorder that had or had not received PMBI. As results, increased FA was demonstrated in the forceps major of subjects with autism spectrum disorder that had received PMBI. Overall, the manuscript is well-written, and the findings appear robust. However, I do have some concerns.

- The major drawback of this study is the small sample size. Furthermore, longitudinal data is preferable to evaluate the effect of treatment/behavioral intervention.

- Why the authors only evaluated FA despite its unspecificity? AD and RD are assumed to represent axon and myelin integrity, respectively, I suggest including MD, AD, and RD in the analysis.

- Were there any significant differences in intelligence and brain volumes between groups? 

- Multiple comparisons correction should be applied in the ROI analysis.

Minor comments:

- This study included not only subjects with autism but also subjects with Asperger syndrome and pervasive developmental disorder; please change the title “autism” —> “autism spectrum disorder.”

- Please use the consistent term “autism spectrum disorder” throughout the manuscript.

- The introduction of this manuscript is too long (4.5 pages) and unfocused.

- Please define all abbreviations on their first use in text, such as DSM, ADOS, and ADIS.

- Page 8, line 176. “Overall, 11 participants received PMBI, while 14 had received other services,” please describe “other services” more precisely and explain will or will not this service affect the results.

- Please provide the p-values in table 1.

- Page 7, line 166: “The final sample of 26 individuals included 17 subjects with ASD, four with Asperger syndrome, and four with pervasive developmental disorder…” Did the total number of participants calculated correctly?

- Please state the duration of PMBI.

- Please describe the statistical analysis of TBSS. Did the authors include age, gender, and intracranial volume as covariates?

6. PLOS authors have the option to publish the peer review history of their article (what does this mean?). If published, this will include your full peer review and any attached files.

Reviewer #1: No

Reviewer #2: **Yes: **Christina Andica

---

## [Author Response · Author response to Decision Letter 0]

17 Dec 2021

IMPORTANT. Submitted also as part of the submission package.

PONE-D-21-30729

Title: A Callosal Biomarker of Behavioral Intervention Outcomes for Autism Spectrum Disorder? A Case-Control Feasibility Study with Diffusion Tensor Imaging

Dear Dr. Bergsland,

Thank you for the opportunity of having our work reviewed at PLOS One. We appreciate your comments and those from the reviewer panel. We believe that these comments have helped to improve the quality of our manuscript very significantly. We are excited to submit a revised manuscript of our study. Below I present a detailed response to all comments. The enclose manuscript has been revised accordingly (edited text in blue fonts). We have made a sincere effort to respond to all concerns as thoroughly as possible (which has required a fresh re-analysis of our dataset). I take the opportunity to thank you for your support throughout the editorial process.

Sincerely,

Javier Virues-Ortega,

On behalf of the authors

Editor comments (different from the reviewers’)

1. TBSS is missing a reference to the original Smith et al. paper. 

This has been corrected. 

2. Please also use the term "sex" rather than "gender", the former which is the biological meaning whereas the latter is the sociological construct, which is not relevant in the context of your manuscript.

Done as suggested.

3. No T1 image is described in the imaging protocol. 

Please, refer to the additions to the Image Acquisition section on p. 11.

4. Finally, please carefully the manuscript for typos (e.g., "Read and green" rather than "Red and green", "mod-to-late" rather than "mid-to-late")

Done as suggested.

Reviewer #1: 

1. The provided data repo link (neurovault.org/collections/ZEPLXWYX , dated +2years ago) does NOT contain the input data for the 30 subjects (nor the 26 used)..only final images.

We have updated the Neurovault file (see link below) to include all datasets (n = 26). All input data will be shared through Figshare upon the manuscript acceptance.

https://neurovault.org/collections/12006/

2. It was said that 30 subject were recruited, one left study, another 4 removed, and 26 remaining! (in this way you started with 31 subjects, or ended with 25 as the case)...in another place, "26 of the 29 subjects passed quality assurance, discarding 3 individuals" implying only 3 had artifacts, and original were 29. Both sentences needs to be altered to be more clear. following, only the age of males were stated ..why? Again, the 26 subjects contained 17 asd, 4Asperger, 4 otherwise (totaling 25?!).

We have reprocessed our data and included one additional dataset. We now provide a more complete narrative of the attrition process (see p. 8). To avoid confusion, we refer to the participant selection process in the Participants section only. 

3. It's also useful to identify which seven has ADHD (all ASD? asperger? which mix?) Especially as they are identified as a separate raw on table1 (raising total count to 32?)

This information has been added to p. 8.

4. It would be beneficial applying the same procedure on a larger publicly available datasets if possible, maybe a subset of Autism Brain Image Data Exchange or others, to confirm generalizability of findings and mitigate the effects of using a small sample size (n<30).

Existing databases (including the ABIDE I and ABIDE II) do not include treatment data. We have indicated in the discussion that a positive addition to these systems may be to include treatment outcome data, maybe following international outcomes standards such as the ICHOM autism coreset.

5. In defining PMBI cases: how long does parent actively conducted those learnt training on their kids? Since this matter directly affect the hypothesis of brain - changes, as some may only conducted those strategies for negligible amount of times. Moreover, what are other services include? and why it's hypothesized it has no effect on the ROI of study, in a way similar to PMBI? Finding an actual distinction can be a result of many other obvious cofounding variables.

Participants received training from qualified therapists for a significant period of time (at least one month). However, training length would not accurately portray the potential impact of the intervention, as parents would have showed different levels of adherence to parent training strategies even if the parent training intervention were comparable in terms of the length of training received by parents. In order to better address these concerns we implemented the following mitigating actions: (1) we now provide a detailed treatment history for exposed individuals and controls (see current Table 1), (2) we classified as exposed individuals those receiving parent training in the context of other treatments (this affected two participants now reclassified as cases that receiving early intensive behavioral intervention, which routinely includes parent training), (3) we replicated the tractography analysis using the total number of distinct interventions received as a covariate, and (4) we conducted a post hoc dose-response analysis using parent training intensity as predictor and FA as outcome. The three levels of parent training intensity are defined as follows.

1 Not receiving parent training (i.e., controls)

2 Receiving parent training not in the context of early intensive behavioral intervention

3 Receiving parent training in the context of early intensive behavioral intervention

Please, refer to the manuscript for the changes described above. 

6. "PMBI can lead to measurable volume changes in known regions of interest for autism": DTI shouldn't reveal volume changes, but micro white matter architecture and integrity.

We have corrected this sentence as suggested.

7. Why only FA was investigated? Not also other common measures as mean diffusivity/ axial diffusivity/ ... ?

We agree with reviewers that FA alone cannot provide a complete picture of tract integrity, and appreciate their dialogue around this issue. Given this project intended to be a proof of concept pilot for future interventional work in ASD, we chose to include the most common DTI metric to describe "tract integrity." To address concerns raised by reviewers we now report axial diffusivity, radial diffusivity, and mean diffusivity for both TBSS and ROI analysis. However, and according to our initial hypothesis, we have kept a targeted ROI analysis focused on FA. However, we now briefly report on the other metrics as well. The following comments has been added to the discussion.

"The current study aimed to explore relationships between common interventions of ASD and white matter integrity. In spite of our tentative findings, it is important to highlight that FA is known to reflect a variety of biological changes within white matter. For example, despite being commonly associated with decreased tract integrity, increased FA could also reflect an increase in neurons, an increase in myelin, or increased inflammation. In order to better characterize what biological mechanism is underpinning the changes we are observing in these brains, future extensions of this work would need to consider mean diffusivity, axial diffusivity, and radial diffusivity in greater detail that it had been possible with the current dataset."

8. The use of references in the discussion doesn't help a lot. For example, Results were said to be consistent with literature, and an example [48] was given although it was anatomical, not WM connectivity (as well as 50).

The paragraph referred by the reviewer has been modified to incorporate a meta-analysis of DTI in autism (p. 15): 

9. Line 143: end of sentence missing (are what?)

Omitted word has been re-added

Reviewer #2: 

Virues-Ortega et al. evaluated white matter integrity between subjects with autism spectrum disorder that had or had not received PMBI. As results, increased FA was demonstrated in the forceps major of subjects with autism spectrum disorder that had received PMBI. Overall, the manuscript is well-written, and the findings appear robust. However, I do have some concerns.

1. The major drawback of this study is the small sample size. Furthermore, longitudinal data is preferable to evaluate the effect of treatment/behavioral intervention.

While the sample size was sufficient for the goals of a feasibility case-control study, and is supported by a post hoc achieved power analysis (beta = 0.94). This has been acknowledged as a key avenue for future research in the discussion (p. 17).

2. Why the authors only evaluated FA despite its unspecificity? AD and RD are assumed to represent axon and myelin integrity, respectively, I suggest including MD, AD, and RD in the analysis.

Please, refer to the response to comment #7 (Reviewer 1).

3. Were there any significant differences in intelligence and brain volumes between groups? 

IQ data was not available. We have added data from an ad hoc autism symptom scale with separate scores for current symptoms and symptoms when first diagnosed. There were no significant differences in autism symptoms at the time of diagnosis (see Table 1). We have also added information on comorbid intellectual disability.

4. Multiple comparisons correction should be applied in the ROI analysis.

We have added the critical p according to Bejamini & Hochberg (1995) multiple-comparison correction. We have revised the ROI to make it more targeted to the corpus callosum in line with our original hypothesis.

5. Minor comments:

- This study included not only subjects with autism but also subjects with Asperger syndrome and pervasive developmental disorder; please change the title “autism” —> “autism spectrum disorder.” Please use the consistent term “autism spectrum disorder” throughout the manuscript.

Done as suggested.

- The introduction of this manuscript is too long (4.5 pages) and unfocused.

We have reduced the length of the introduction to 3 pages. We have made a serious attempt to make the introduction focused and succinct. 

- Please define all abbreviations on their first use in text, such as DSM, ADOS, and ADIS.

Done as suggested.

- Page 8, line 176. “Overall, 11 participants received PMBI, while 14 had received other services,” please describe “other services” more precisely and explain will or will not this service affect the results.

Please, refer to the answer to Reviewer #1 (Comment 5) for an in-depth response to this concern. The current Table 1 includes full details of other interventions (see also edits in p. 9).

- Please provide the p-values in table 1.

Done as suggested.

- Page 7, line 166: “The final sample of 26 individuals included 17 subjects with ASD, four with Asperger syndrome, and four with pervasive developmental disorder…” Did the total number of participants calculated correctly?

Please, refer to the answer to Reviewer #1 (Comment 2).

- Please state the duration of PMBI.

Please, refer to the answer to Reviewer #1 (Comment 2).

- Please describe the statistical analysis of TBSS. Did the authors include age, gender, and intracranial volume as covariates?

We included age and gender as covariates. Intracranial volume was not statistically different across groups and was not included as a regressor (see p. 13).

---

## [Decision Letter · Decision Letter 1]

30 Dec 2021

A Callosal Biomarker of Behavioral Intervention Outcomes for Autism Spectrum Disorder? A Case-Control Feasibility Study with Diffusion Tensor Imaging

PONE-D-21-30729R1

Dear Dr. Virues-Ortega,

We’re pleased to inform you that your manuscript has been judged scientifically suitable for publication and will be formally accepted for publication once it meets all outstanding technical requirements.

Kind regards,

Niels Bergsland

Academic Editor

PLOS ONE

Additional Editor Comments (optional):

Reviewers' comments:

Reviewer's Responses to Questions

**Comments to the Author**

1. If the authors have adequately addressed your comments raised in a previous round of review and you feel that this manuscript is now acceptable for publication, you may indicate that here to bypass the “Comments to the Author” section, enter your conflict of interest statement in the “Confidential to Editor” section, and submit your "Accept" recommendation.

Reviewer #2: All comments have been addressed

2. Is the manuscript technically sound, and do the data support the conclusions?

Reviewer #2: Yes

3. Has the statistical analysis been performed appropriately and rigorously? 

Reviewer #2: Yes

4. Have the authors made all data underlying the findings in their manuscript fully available?

Reviewer #2: Yes

5. Is the manuscript presented in an intelligible fashion and written in standard English?

Reviewer #2: Yes

6. Review Comments to the Author

Reviewer #2: The authors have satisfactorily responded to all my comments and suggestion. This manuscript is now acceptable for publication.

7. PLOS authors have the option to publish the peer review history of their article (what does this mean?). If published, this will include your full peer review and any attached files.

Reviewer #2: **Yes: **Christina Andica

---

## [Editor Report · Acceptance letter]

19 Jan 2022

PONE-D-21-30729R1 

A Callosal Biomarker of Behavioral Intervention Outcomes for Autism Spectrum Disorder? A Case-Control Feasibility Study with Diffusion Tensor Imaging 

Dear Dr. Virues-Ortega:

I'm pleased to inform you that your manuscript has been deemed suitable for publication in PLOS ONE. Congratulations! Your manuscript is now with our production department. 

Kind regards, 

on behalf of

Dr. Niels Bergsland 

Academic Editor

PLOS ONE